# Genetic, Transcriptomic, and Epigenomic Insights into Sjögren’s Disease: An Integrative Network Investigation and Immune Diseases Comparison

**DOI:** 10.3390/ijms26104637

**Published:** 2025-05-13

**Authors:** Nitesh Enduru, Astrid M. Manuel, Zhongming Zhao

**Affiliations:** 1Center for Precision Health, McWilliams School of Biomedical Informatics, The University of Texas Health Science Center at Houston, Houston, TX 77030, USA; nitesh.enduru@uth.tmc.edu (N.E.); astrid.manuel@bcm.edu (A.M.M.); 2Department of Epidemiology, Human Genetics and Environmental Sciences, School of Public Health, The University of Texas Health Science Center at Houston, Houston, TX 77030, USA; 3Molecular & Human Genetics Department, Baylor College of Medicine, Houston, TX 77030, USA

**Keywords:** autoimmune disease, genome-wide association studies, pleiotropy, Mendelian randomization, genetic correlation, drug target

## Abstract

Sjögren’s disease (SjD) is a systemic autoimmune disorder primarily causing dry eyes and mouth. It frequently overlaps with other autoimmune diseases (AIDs), including rheumatoid arthritis (RA) and systemic lupus erythematosus (SLE). However, the genetic basis of SjD remains underexplored, limiting our understanding of its connections to other immune-mediated conditions. In this study, we aimed to identify gene networks associated with SjD through the integration of genetic, transcriptomic, and epigenomic data. We further compared the genetic factors of SjD with other immune-mediated diseases. We analyzed genome-wide association studies (GWAS) summary statistics, DNA methylation, and transcriptomic data using our in-house network-based methods, dmGWAS and EW_dmGWAS, to identify key gene modules associated with SjD. In dmGWAS analysis, discovery and evaluation datasets were used to identify consensus results. We conducted gene-set, cell-type, and disease-enrichment analyses on significant gene modules and explored potential drug targets. Genetic correlations and Mendelian randomization were applied to assess SjD’s link with 17 other AIDs and 16 cancer types. dmGWAS identified 207 and 211 gene modules in the discovery and evaluation phases, respectively, while EW_dmGWAS detected 886 modules. Key modules highlighted 55 genes (discovery), 52 genes (evaluation), and 59 genes (EW_dmGWAS), with at least 50 genes from each analysis linked to AIDs and cancer. Enrichment analyses confirmed their relevance to immune and oncogenic pathways. We pinpointed four candidate drug targets associated with AIDs. We developed a novel integrative omics approach to identify potential genetic markers of SjD and compared them with AIDs and cancers. Our approach can be similarly applied to other disease studies.

## 1. Introduction

Sjögren’s disease (SjD) is a chronic autoimmune disease (AID) primarily affecting the endocrine glands, leading to dry mouth and eyes due to dysfunction of salivary and lacrimal glands [1,2,3]. Beyond glandular involvement, SjD can manifest systemically, impacting the joints, lymphatic system, nervous system, lungs, and kidneys [1,2]. Despite its clinical significance, particularly among women, the exact etiology remains elusive, with genetic and environmental factors playing crucial roles [4].

Genetic predisposition to SjD is supported by familial aggregation studies and twin registries, highlighting a heritable component [5]. Less than 100 genetic variants associated with SjD have been identified through genome-wide association studies (GWAS), implicating genes involved in modulating immune responses and increasing susceptibility to autoimmune conditions [6]. Notably, several variants reside in genes related to the major histocompatibility complex (MHC) and non-MHC genes involved in immune regulation and epithelial function [7]. Histopathologically, SjD is characterized by lymphocytic infiltration of exocrine glands, primarily by activated CD4+ T cells, which produce proinflammatory cytokines such as tumor necrosis factor-α (TNF-α) and interleukins (IL)-1β, IL-2, IL-4, and IL-6 [8,9]. However, the precise molecular mechanisms driving disease onset and progression remain unclear.

Recent advancements in genomics and multi-omics technologies provide novel opportunities to dissect the complex genetic and molecular landscape of AIDs [10]. Despite the identification of several genetic risk loci associated with SjD, translating these findings into practical biological insights has remained challenging [6]. Integrating multi-omics data, incorporating genomic, epigenomic, transcriptomic, and proteomic layers, can offer deeper insights into disease mechanisms. By leveraging bioinformatics tools, we constructed biological networks to uncover key regulatory nodes and functional pathways.

In this study, we employed our in-house network-based approaches, dense module search of GWAS (dmGWAS) and edge-weighted dmGWAS (EW_dmGWAS), to bridge the gap between genetic susceptibility and disease manifestation. Using GWAS summary statistics, epigenomics, transcriptomics, and proteomics data, we aimed to identify genetic networks associated with SjD, elucidate their functional roles, and explore their links to other autoimmune diseases. This integrative framework provides a systems-level understanding of SjD, with potential implications for biomarker discovery and targeted therapies.

## 2. Results

The overall workflow of the present study is summarized in Figure 1. We integrated genetic, epigenomic, transcriptomic, and proteomic data to investigate genetic signals associated with SjD at the network and pathway levels. Further, we explored the genetic relationship of SjD with cancer and other AIDs.

### 2.1. Identification of SjD-Associated Genes Using dmGWAS

To identify Sjögren’s syndrome (SjD)-associated gene modules, we integrated GWAS summary statistics with epigenomic data from minor salivary glands (MSGs) using dmGWAS. Genetic and epigenomic weights were combined using a scaling factor to generate node weights, as described in Section 4. The larger dataset was designated as the discovery set, while the smaller dataset served as the evaluation set for validation (Appendix A).

Genetic weights were derived from MAGMA-based gene annotation of GWAS SNPs (18,716 genes), while epigenomic weights were calculated from differentially methylated probes (Appendix A). After integrating the genetic and epigenomic weights (known as node weights), we performed a dense module search on 15,664 unique genes in the discovery data using dmGWAS. We identified 207 significant modules with 1821 unique genes (module score (Z_m_) > 1.96) (Appendix A). The results of the evaluation dataset showed 211 gene modules with 1998 unique genes in the validation data (Appendix A) and had a module score (Z_m_) > 1.96. These results highlight robust and reproducible gene network modules associated with SjD, providing insights into potential regulatory mechanisms underlying disease pathogenesis.

### 2.2. Autoimmune and Cancer-Related Genes in the Top Gene Modules

For the discovery data, the top module with the highest score consisted of 10 genes (*BLK*, *CDC37*, *DDX6*, *FAM167A*, *MAPT*, *STAT1*, *STAT4*, *TNIP1*, *TNPO3*, *UBE3A*) (Appendix A). Six out of these ten genes were previously identified as GWAS significant (*BLK*, *MAPT*, *STAT1*, *STAT4*, *TNIP1*, *TNOP3*) [11]. The genes *BLK*, *FAM167A*, *STAT1*, *STAT4*, *TNPO3*, and *TNIP1* are associated with AIDs, and *CDC37*, *DDX6*, *MAPT*, *STAT1*, *STAT4*, and *UBE3A* are associated with cancers [11]. We further examined the top five modules and identified 23 SjD candidate genes (Appendix A). Further, we selected a final subnetwork consisting of the top 24 modules with 55 genes based on the R2 cutoff for further analysis (see Section 4) (Figure 2A and Appendix A).

For the evaluation data, the top module consisted of 12 genes (*BLK*, *CD247*, *CDC37*, *CSK*, *DDX6*, *HSPA6*, *IRF5*, *MAPT*, *PPIB*, *STAT1*, *STAT4*, *TNIP1*) (Appendix A). Seven out of twelve genes were previously identified as GWAS significant (*BLK*, *CD247*, *MAPT*, *STAT1*, *STAT4*, *TNIP1*, and *TNOP3*). The genes *BLK*, *CDC247*, *CSK*, *IRF5*, *STAT1*, *STAT4*, and *TNIP1* are associated with AIDs, and *CDC37*, *DDX6*, *HSPA6*, *MAPT*, *PPIB*, *STAT1*, and *STAT4* are associated with cancers [11]. Seven genes (*BLK*, *CDC37*, *DDX6*, *MAPT*, *STAT1*, *STAT4*, and *TNIP1*) from the top one module were shared between the discovery and evaluation data. We further examined the top five modules and identified 30 SjD candidate genes (Appendix A). Further, we selected a final subnetwork consisting of the top 13 modules with 52 DMGs based on the R2 cutoff for further analysis (Figure 2B and Appendix A). These findings reinforce the genetic links between SjD, other autoimmune diseases, and cancer, highlighting key regulatory genes that may drive disease susceptibility and progression.

### 2.3. Overlap Between the Discovery and Evaluation Data

We identified 25 TMGs overlapping between the discovery and evaluation data from the final subnetwork modules. Of these, 18 genes exhibited significant node weight (|Z| > 1.96) (Appendix A). Among the 18 genes, 11 of them were either hypomethylated or hypermethylated in both discovery and evaluation data. In contrast, seven of them had the opposite direction of methylation in discovery and evaluation data (Appendix A).

### 2.4. EW_dmGWAS Identified a Surplus of Gene Network Modules

Using SjD GWAS summary statistics and MSG transcriptomic data, the EW_dmGWAS (version 3.1) model discovered TMGs related to AIDs and cancer. This method integrates GWAS-derived node weights with transcriptomic-derived edge weights, allowing for a network-based identification of disease-relevant gene modules. We condensed the SjD GWAS summary statistics to 18,715 adjacent genes using the MAGMA tool (v1.10). The gene node weight value was then transformed from gene-level significance (see Section 4) (Appendix A). Second, edge weights were determined using MSG-specific transcriptomic data, including 10,647 gene-level transcriptomic data (Appendix A). Each dataset was subjected to data normalization, batch correction, and missing data imputation. The edge weights were then estimated using the differential co-expression profiles of Sjogren’s cases and control samples. These values were then matched to PPIs from the BioGRID database (see Section 4).

Following data harmonization, 10,646 gene node weights and 340,151 edge weights were employed in the EW_dmGWAS analysis (Appendix A). To balance the impact of genetic variations and human protein interactomes, we established a scaling factor (λ = 1.22), calculated by the ratio of node and edge weight variances. Finally, we discovered 923 dense gene modules using MSG tissue’s gene-level protein expression information, and 886 of them had module scores (Zm) > 1.96 (Appendix A).

EW_dmGWAS identified 923 modules with 1537 unique genes. Four genes (BRCA1, PSMD4, RPL3, STAT1) were identified in the top one module (Appendix A). Three of these four genes, BRCA1, PSMD4, and RPL3, are reported as cancer genes, and STAT1 is associated with both cancer and AIDs [12]. We further examined the top five modules and identified 11 SjD candidate genes (Appendix A). Further, we selected a final subnetwork within the top 10% (33 modules with 59 DMGs) based on the R2 cutoff for further analysis (Figure 2C and Appendix A) (see Section 4). These findings underscore the utility of EW_dmGWAS in identifying gene networks with cross-disease relevance, reinforcing potential shared molecular mechanisms between SjD, autoimmune diseases, and cancer.

### 2.5. Selection and Visualization of TMGs from dmGWAS and EW_dmGWAS

We prioritized SjD-associated dense modules using 1000-permutation z-scores and examined the network features of genes inside an increasing number of top modules. To evaluate the network characteristic, we estimated the coefficient of determination of the log–log relationship of edge frequency over nodes, R2, which measures the degree of scale-free networks. As illustrated in Appendix A, the scale-free network indexes are positively connected with the number of modules chosen at the outset, and the index stabilized as more modules were combined. Based on the R2 values, we empirically selected the top 10% (top 24 modules from the discovery data, top 13 modules from the evaluation data for dmGWAS analysis, and top 33 modules for EW_dmGWAS analysis (R2 = 0.8)) as a cutoff threshold for visualization and further evaluation. The median scale-free network index R2 was 0.77 for the top 24 discovery gene modules with 55 genes and 0.79 for the top 13 evaluation gene modules with 52 genes, showing almost scale-free networks for both outcomes. For the transcriptomic data, the median scale-free network index R2 was 0.82 for the top 33 modules with 59 genes.

### 2.6. Overlap Between dmGWAS and EW_dmGWAS Findings

Overall, dmGWAS identified 372 and 415 genes across the discovery and evaluation data, and 245 genes overlapped among them. EW_dmGWAS identified 1537 genes. Among the dmGWAS and EW_dmGWAS, 173 genes overlapped. Among the top subset dmGWAS modules, 25 genes overlapped across the discovery and evaluation data (Appendix A). A total of 18 out of the 25 genes were statistically significant for hypo/hypermethylation. Seven out of eighteen genes had opposing methylation effects in discovery and evaluation data. Between the dmGWAS discovery and EW_dmGWAS modules, five genes overlapped (*JARID2*, *PPARD*, *RPL3*, *STAT1*, and *TYK2*). Between the dmGWAS discovery, evaluation, and EW_dmGWAS data, three genes overlapped (*PPARD*, *RPL3*, and *STAT1*).

### 2.7. Exploration of Biological Mechanisms: Gene, Disease, and Cell Specificity Enrichment Analyses

To investigate the functional and disease relevance of top module genes (TMGs), we performed gene ontology (GO), disease enrichment, and cell-type specificity analyses. We conducted gene and disease enrichment analyses using the Cluster Profiler R package (v4.2.2). We used GO annotation terms to perform gene over-representation analyses (ORAs). We used discovery and evaluation data of the final subset of modules for this analysis. For both modules, similar pathways were enriched, and multiple enriched biological process pathways were previously associated with SjD. Pathways related to the growth hormone receptor signaling pathway via JAK-STAT (*p*-value = 2.9 × 10^−4^), regulation of bone remodeling (*p*-value = 3.1 × 10^−4^), type II interferon-mediated signaling pathway (*p*-value = 4.9 × 10^−5^), cellular response to type II interferon (*p*-value = 2.6 × 10^−4^) and interleukin-12 (*p*-value = 3.6 × 10^−4^), and negative regulation of catabolic process (*p*-value = 4 × 10^−4^) were among the top enriched pathways in the GO BP domain (Figure 3A,B). For the final subset of modules, pathways related to cytoplasmic translation (*p*-value = 5.7 × 10^−7^), regulation of inflammatory response (*p*-value = 3.1 × 10^−4^), epigenomic regulation of transcriptomic (*p*-value = 5.7 × 10^−5^), and erythrocyte homoeostasis (*p*-value = 7.9 × 10^−5^) were found (Figure 3C).

We used Disease Ontology terms to perform disease ORA. The TMGs were associated with AIDs such as SLE, herpes simplex, rheumatic–psoriatic arthritis (RA/PA), and others in discovery and evaluation data (Figure 3D,E). The TMGs from transcriptomic data were associated with different anemia diseases (Figure 3F).

To characterize the tissue and cell types across the TMGs, we used the WebCSEA tool (https://bioinfo.uth.edu/webcsea/, accessed on 28 October 2024). There was significant enrichment of classical monocyte, endothelial cell, cd8-positive, and alpha–beta T cell types across blood, skin, muscle, and thymus tissues (Appendix A) in both top modules. For the TMGs, monocytes, granulocytes, B cells, and T cells were enriched across the blood, liver, and small intestine tissues (Appendix A).

### 2.8. Proposing Repurposable Drug Candidates for SjD

The drug target enrichment analysis of our TMGs revealed 31 genes encoding drug targets associated with various cancers and autoimmune diseases (Appendix A). These drug targets were annotated by the Therapeutic Target Database (TTD) to pertain to four categories: ‘successful targets’, ‘clinical trial targets’, ‘patented recorded targets’, and ‘literature-reported targets’. We focused on the ‘successful targets’, as these have corresponding FDA-approved medications, which serve as potential repurposable candidates for Sjogren’s disease. Figure 3G summarizes these approved drugs, their corresponding targets, and the current indications of these medications. Notably, Phenyltoloxamine, an antihistamine currently indicated for allergic disorders, is a promising repurposable drug candidate for SjD. Similarly, Acitretin and Deucravacitinib, both approved for the treatment of psoriasis, are also suggested as repurposable candidates for SjD. Moreover, we suggest two cancer drugs as potential repurposable candidates for SjD: Tucatinib and Secretin. These findings highlight potential drug repurposing opportunities for targeted SjD therapies, emphasizing the therapeutic relevance of immune-modulating and anti-inflammatory drugs in SjD treatment.

### 2.9. Genetic Correlation with Cancer and Other AIDs

Previous studies suggested a strong genetic correlation (rg) of SjD with other AIDs. We further explored the rg between SjD and other AIDs related to skin disorders, cancers, and eye diseases, as well as with aging diseases like Alzheimer’s disease (AD) and frailty. We calculated the rg between SjD and 17 other AIDs. Except for Crohn’s disease (CD) and ulcerative colitis (UC), SjD was relatively correlated with other AIDs (Figure 4A, Appendix A). Among the 17 cancers, SjD had a strong correlation with non-Hodgkin’s lymphoma, oralpharynx, and thyroid cancers (Figure 4B, Appendix A). Among the eye diseases, only early age-related macular degeneration had a relative correlation with SjD (Appendix A). Among the aging diseases, SjD was correlated with the frailty index (FI) (Appendix A).

### 2.10. Causal Relationship with Cancer and Other AIDs

Given the observed genetic correlations, we further explored the causal relationship of SjD with other diseases mentioned above. We evaluated the potential causal relationship between SjD and AIDs, cancer, eye diseases, and other aging diseases using the inverse-variance weighting (IVW) implemented in the 2SMR package (v0.6.5). Among the AIDs, there was a significant causal relation of SjD with PA, RA, and SLE (Figure 5A). Among the cancers, only ovarian cancer was causally associated with SjD (Figure 5B). SjD showed a causal relationship between high myopia (HM), among the ocular diseases (Appendix A), and frailty phenotype, among the aging diseases (Appendix A).

## 3. Discussion

This study employed a comprehensive multi-omics, network-based approach using dmGWAS and EW_dmGWAS frameworks to investigate the genetic, epigenomic, and transcriptomic landscape of SjD. By integrating GWAS, DNA methylation, and transcriptomic data, we identified key gene modules associated with autoimmune pathways and cancer-related processes. Our findings provide new insights into the molecular mechanisms of SjD and its potential genetic links to other diseases.

The dmGWAS analysis revealed gene modules enriched in genetic and epigenomic alterations associated with SjD, many of which were involved in immune response pathways such as interferon signaling, a key driver of autoimmune processes. Several top-ranking genes, including *BLK*, *STAT4*, and *TNIP1*, have been previously implicated in SjD and other AIDs [13,14,15], validating our approach. Beyond autoimmunity, our analysis identified genes with dual roles in AIDs and cancer, such as *CDC37*, *DDX6*, and *MAPT* [16,17,18], suggesting possible immune–oncogenic links. Notably, *STAT1*, which was identified in both dmGWAS and EW_dmGWAS, is a key regulator in both autoimmune diseases and cancer progression [19,20,21].

Our EW_dmGWAS analysis, integrating GWAS with transcriptomic data, expanded our understanding by identifying differential co-expression profiles linked to SjD. The genes identified through EW_dmGWAS significantly overlapped with known AID-associated genes, highlighting their potential cross-disease relevance. Specifically, genes like *PPARD*, *RPL3*, and *STAT1*, which overlapped in both our epigenomic and transcriptomic analyses, are highly interconnected across AIDs and cancer [21,22,23,24].

Pathway enrichment analyses highlighted immune and inflammatory mechanisms as key contributors to SjD pathogenesis. Enrichment analyses in these modules pointed to biological processes like cytoplasmic translation and the regulation of inflammatory responses. These could represent mechanistic links between genetic predisposition and the inflammatory manifestations seen in SjD [25]. The pathways of cellular response to type II interferon play a vital role in the development and severity of AIDs like SLE [26] and are considered potentially helpful for adjuvant immunotherapy for different types of cancer [27]. The alpha–beta T cell activation is critical in immune defense and is often dysregulated in autoimmunity. In cancer, T cell activation can be beneficial for tumor suppression but can also be a target for tumors to evade immune responses [28]. The overlap of these pathways across disease states highlights shared molecular mechanisms that may contribute to both autoimmunity and tumorigenesis [29,30,31,32,33].

We expanded our analysis to disease enrichment, where we found associations between SjD modules and conditions such as RA, SLE, and other diseases. Our drug target enrichment analysis identified FDA-approved medications as potential repurposable therapies for SjD, including Tucatinib and Deucravacitinib, offering the potential for therapeutic repurposing [34]. Drugs targeting pathways such as the JAK-STAT signaling axis may provide benefits in SjD, given the observed enrichment of these pathways [35]. Moreover, our genetic correlation analyses identified significant associations between SjD and specific cancers, including non-Hodgkin’s lymphoma and thyroid cancer [36], reinforcing epidemiological evidence of increased malignancy risk in SjD patients. These findings underscore the importance of cancer surveillance in SjD patients and warrant further research into shared genetic susceptibilities. We further explored if the TMGs of SjD are associated with the antiphospholipid antibody syndrome (APS) pathways. The presence of antiphospholipid antibodies is part of the diagnostic criteria for several rheumatic diseases, including Sjogren’s disease, and often leads to serious thrombotic events [37,38]. Several TMGs have been implicated in immune regulation, signaling, and cellular stress pathways relevant to APS, including *BLK*, *CDC37*, *DDX6*, *FAM167A*, *STAT1*, and *STAT4*. Among them, *STAT1* and *STAT4* are directly involved in interferon and cytokine signaling, which are central to APS-related inflammation and autoimmunity [29,30]. *BLK* and *FAM167A* influence B cell function and autoantibody production, contributing to the generation of pathogenic antiphospholipid antibodies [30,31]. *DDX6* and *CDC37* regulate cellular stress responses and autophagy, processes that can impact endothelial function and thrombosis in APS [32,33]. So far, *MAPT*, *TNIP1*, *TNPO3*, and *UBE3A* have less clearly defined roles in APS.

AIDs exhibit notable sex-based dimorphism, with a higher prevalence in females. This sex bias is particularly evident in SjD, APS, and RA [39]. Male and female patients with SjD demonstrate distinct disease manifestations. SjD, for instance, has a striking female predominance, with female-to-male ratios ranging from 6:1 to 16:1, particularly in older women post-menopause [12]. APS also shows a female bias, with women typically developing the disease earlier and experiencing more venous thromboses, while men are more prone to arterial events [40]. RA affects women 2–3 times more often than men, with women often experiencing more severe symptoms and poorer treatment outcomes [41]. Females with SjD are at higher risk for conditions like fibromyalgia, migraines, and Ehlers–Danlos syndrome, while males have increased risks of cerebrovascular and cardiac issues. However, the relationship between these comorbidities and SjD itself remains unclear [12]. These sex differences are believed to arise from complex interactions between genetics, sex hormones, and environmental factors, highlighting the need for sex-specific approaches in both research and treatment.

Further, the X chromosome plays a key role in the sex differences seen in SjD, as many immune response genes are located on the X chromosome. Females, with two X chromosomes, may be more prone to immune dysregulation than males, who have only one X chromosome [42]. A notable study found that women with trisomy X have a 2.9-fold higher risk of developing primary Sjögren’s disease compared to women with normal XX karyotypes [39]. X chromosome aneuploidies, such as Klinefelter syndrome (47,XXY) in men and triple X syndrome (47,XXX) in women, are significantly enriched among patients with Sjögren’s syndrome compared to the general population, supporting the concept of an X chromosome dose effect in disease susceptibility [43,44]. This increased risk is thought to be mediated by immune-related genes on the X chromosome, such as *TLR7* and *CXorf21*, that escape X inactivation, leading to overexpression and heightened immune activation, which helps explain the strong female predominance observed in Sjögren’s syndrome [45]. Many of the significant genetic associations related to SjD are concentrated in the MHC region and vary depending on the patient’s serological status [42]. This highlights the complex interplay between genetic factors, X chromosome dosage, and immune response in the development of SjD, suggesting that the female bias in this disease may be partly driven by X-linked gene expression. Currently, there has been no GWAS study specifically focused on sex differences in SjD due to limited sample size in the existing studies.

Despite the insights and potential genetic markers identified from this multi-omics network study, it has several limitations. First, there were limited available data for this study; therefore, the sample size and tissue sources may introduce variability. Second, this study focuses on MSG tissues to represent the molecular landscape of SjD, limiting insight into other affected tissues and systemic manifestations. Third, our multi-omics analysis, which integrated GWAS with transcriptomic and epigenetic data, was conducted using the datasets derived solely from females. This lack of gender diversity may limit the generalizability of our findings across both sexes. Future studies incorporating male data would be valuable to assess potential sex-specific differences in the molecular mechanisms underlying SjD. While integrative analyses identified gene modules and drug targets associated with SjD, these associations do not establish causation, and further functional studies are needed to confirm their roles. The identified drug targets also require experimental validation to assess their therapeutic potential for SjD. Finally, our study’s focus on individuals of European ancestry may limit its applicability across other populations. Expanding research to include diverse groups could reveal population-specific genetic variations, enhancing the understanding of SjD.

## 4. Materials and Methods

### 4.1. Multi-Omics Data Overview

#### 4.1.1. Compilation of Sjögren’s Specific Multi-Omics Data

We accessed GWAS summary statistics from the database of Genotypes and Phenotypes (dbGaP) (accession number: phs002723.v1.p1). The original study, conducted by Khatri et al. [6], included 3232 SjD cases and 17,481 controls of European ancestry.

#### 4.1.2. DNA Methylation Data Retrieval

Publicly available DNA methylation data from minor salivary glands (MSGs) of Sjögren’s syndrome (SjD) cases and controls were obtained from the Gene Expression Omnibus (GEO) database (http://www.ncbi.nlm.nih.gov/geo, accessed on 15 November 2024). Two datasets were selected for analysis: a discovery dataset (GSE110007) and an evaluation dataset (GSE166373), both generated using the Illumina HumanMethylation450 BeadChip platform.

The discovery dataset (GSE110007) comprised 13 SjD cases and 15 controls [46]. The evaluation dataset (GSE166373) included methylation profiles obtained from both the Illumina HumanMethylation450 BeadChip and Infinium Methylation EPIC BeadChip platforms. To ensure consistency with the discovery dataset, only the 13 cases and 15 controls profiled using the HumanMethylation450 BeadChip were included in the analysis [47]. Both datasets were generated through the Sjögren’s International Collaborative Clinical Alliance (SICCA) Registry.

#### 4.1.3. Gene Expression Data and Analysis

We obtained the transcriptomic data acquired from MSG samples (9 SjD cases, 8 controls) available under GEO accession ID GSE157159. This dataset was generated at the Oklahoma Medical Research Foundation using the Illumina NovaSeq 6000 platform [48]. Further technical details regarding RNA extraction and sequencing procedures can be found in the original study [48].

#### 4.1.4. Protein–Protein Interaction Network

We retrieved a reference human protein–protein interaction (PPI) network to perform dense module searching for gene network modules associated with SjD. We used PPI data from the BioGRID database (version 4.4.203), a publicly accessible comprehensive collection of PPI interactions that is available through the BioGRID platform [49]. We filtered the experimentally validated human interactions.

### 4.2. Gene Network Analysis

#### 4.2.1. dmGWAS Analysis: Discovery and Evaluation of Integrated Genetic and Epigenomic Data

To identify SjD-associated gene network modules, we employed the dense module search of GWAS (dmGWAS), a statistical framework designed to integrate genetic and epigenomic data. dmGWAS utilizes a greedy search algorithm to detect highly enriched gene modules within a PPI network, incorporating signals from GWAS and DNA methylation datasets [50,51]. To ensure the reproducibility of identified gene modules, we implemented a discovery and evaluation framework, independently analyzing two DNA methylation datasets.

In the discovery phase, we integrated the epigenomic data (GSE110007) with GWAS summary statistics using the dmGWAS framework to identify gene modules associated with SjD. To this end, we mapped methylation intensities to gene loci and integrated these with GWAS signals. The dmGWAS algorithm was applied to the PPI network, where nodes were weighted based on the gene-level associations derived from GWAS and methylation data. The GSE166373 dataset served as an evaluation dataset to validate the modules identified in the discovery phase. By replicating the dmGWAS analysis with this independent epigenomic dataset, we assessed the consistency and robustness of the identified gene modules.

#### 4.2.2. Dense Module Search on Integrated Genetic and Epigenomic Data

We applied the dmGWAS (version 2.7) [52] to identify gene modules associated with SjD by integrating genetic and epigenomic node weights with a human PPI network. The dense module searching (DMS) algorithm was used for this analysis. Briefly, DMS starts by selecting a random seed gene from the network, which serves as a starting point for the expansion of gene modules. The algorithm iteratively recruits neighboring genes (genes within a predefined distance, d, from the seed) that improve the module score (Zm), based on gene-level z-scores. Each new gene added to the module results in a recalculated module score (Zm + 1). This process continues until no further improvement greater than Zm × r is observed. The parameter r controls the expansion threshold: a smaller r value (e.g., r = 0.05) results in larger modules, while a larger value (e.g., r = 0.2) imposes stricter limits on expansion. We used the default parameters (d = 2 and r = 0.1), as recommended by the dmGWAS user guide [52].

To convert DNA methylation data into gene-level scores, we processed CpG-level methylation intensities using the Limma R package (v3.50.3) [53,54]. Gene-level methylation scores were computed using Stouffer’s Z-score method, which aggregates methylation effects at transcription start sites (TSSs), capturing regulatory regions 1500 base pairs upstream [50,55]. To apply this, we first annotated the CpGs to the TSS of genes. The genomic coordinates of the CpG sites were mapped to the TSS regions of the candidate genes utilizing annotations defined in the Illumina Human Methylation 450 BeadChip annotation file. This method further allowed us to know the directionality of the methylation; a positive Z-score indicated hypermethylation, and a negative Z-score indicated hypomethylation. A detailed explanation of this process is described [50].

To calculate node weights of SjD GWAS, we used MAGMA (multi-marker analysis of genomic annotation), a gene-level analysis method based on GWAS data that calculates *p*-values for genes [56]. SNP-to-gene annotation was performed using a gene boundary window of 35 kb upstream and 10 kb downstream of the gene loci. We then calculated the node weights as z-scores from the gene-level *p*-values using the inverse normal distribution function. The MHC region (chr6:25,000,000–33,500,000) was excluded due to its high linkage disequilibrium (LD) association with Sjögren’s syndrome [4].

To integrate genetic (GWAS-based) and epigenomic (methylation-based) scores, we calculated a scaling factor based on the variance ratio between GWAS and methylation z-scores, ensuring comparable contributions from both datasets before module detection. This integrative network-based approach facilitated the identification of key genetic modules associated with SjD, enhancing our understanding of epigenetic regulation in autoimmune disease susceptibility.

#### 4.2.3. Edge-Weighted Dense Modules Search on Integrated Genetic and Transcriptomic Data

To identify gene modules associated with SjD, we employed EW_dmGWAS software (version 3.1) [57], an extended network-assisted algorithm based on dmGWAS. This approach integrates GWAS-derived node weights and transcriptomic-derived edge weights, enabling the identification of dense gene modules enriched for SjD-related molecular interactions. The EW_dmGWAS algorithm employs a greedy search strategy to detect highly connected gene modules, calculating module scores based on their enrichment in genetic and transcriptomic signals [50,51]. To balance the contributions of GWAS and transcriptomic data, we applied a scaling factor, determined by the variance ratio between edge weights and node weights. To evaluate the significance of the detected modules, we performed 1000 permutations, generating a normalized module score distribution for comparison.

The module search process in EW_dmGWAS follows a similar approach to that in the original dmGWAS algorithm. To assess the significance of the identified modules, we calculated a normalized module score based on the distribution of module scores from a subset of modules over 1000 permutations.

Node weights of EW_dmGWAS were calculated using MAGMA, similar to that of node weight calculation of dmGWAS. We used pre-processed transcriptomic data from MSG to compute edge weights between genes. Edge weights were determined by analyzing the changes in gene co-expression between SjD cases and controls. For each gene pair, we calculated Pearson’s correlation coefficients based on the transcriptomic data from both cases and controls. To assess the significance of differences in co-expression between these groups, we applied Fisher’s transformation and Fisher’s test.

Next, gene pairings were mapped onto two reference PPI databases: an empirically validated PPI database and the BioGRID. After removing redundant and non-human interactions, the final dataset included 19,094 genes and 539,890 unique human PPIs. By integrating processed transcriptomic data with the curated PPI network, we generated edge weights reflecting differential co-expression profiles between SjD cases and controls.

We defined differential co-expression based on edge weight changes that exceeded the nominal significance threshold (edge weight > 1.96), indicating substantial changes in co-expression. This edge-weighted network-based approach allowed for the identification of disease-relevant molecular interactions, providing insights into gene regulatory mechanisms underlying SjD. Further methodological details can be found in the original publication [57].

#### 4.2.4. Module Networks Evaluation and Selection of Top Modules

To identify biologically significant gene modules, we evaluated the top modules of dmGWAS- and EW_dmGWAS-derived networks using a scale-free network analysis. First, all modules were ranked based on the 1000-permutation z-scores calculated in dmGWAS and EW_dmGWAS. Second, we selected the top modules and assessed their scale-free network properties. Specifically, we curated all genes within the selected modules and reconstructed the merged network based on the BioGRID PPI. Scale-free network indexes were calculated and plotted to assess the network properties with the increasing size of the network. We assume the distribution of nodes and frequency of edges within a reconstructed biological network follow the power law, P(k) ∼ k − γ, where k stands for the number of node edges [58]. After applying the log–log transformation, a linear fit, log(pk) = −γ log(k) + c, can be found, and the coefficient of determination of the regression (R2) can be used to estimate the scale-free network property. Based on R^2^ values, we prioritized high-confidence modules (R^2^ ≈ 0.8) for each of the epigenomic- and transcriptomic-based modules. These top-ranked modules were selected for visualization, biological function exploration, and downstream pathway analysis, ensuring that the identified networks represented robust and functionally relevant interactions.

### 4.3. Functional Enrichment Analysis

#### 4.3.1. Gene Ontology and Disease Enrichment

To assess the biological functions and disease associations of the top module genes (TMGs) identified in dmGWAS and EW_dmGWAS analyses, we performed over-representation analysis (ORA) and disease enrichment analysis (DEA) using the clusterProfiler R package (V4.14.4) [59]. The analysis was conducted based on human-specific annotations from the org.Hs.eg.db database.

We used the enrichGO function in clusterProfiler, applying the hypergeometric test to determine statistical significance. To account for multiple comparisons, *p*-values were adjusted using the Benjamini–Hochberg method, with a significance threshold of FDR < 0.1. The top ten enriched GO-BP terms were visualized using dot plots to highlight the most relevant biological processes associated with SjD.

To investigate disease associations, we used the enrichDO function to perform disease enrichment analysis based on Disease Ontology (DO) annotations. Statistical significance was assessed using a hypergeometric test (*p*-value < 0.05 cutoff). The most significantly enriched disease terms, along with their *p*-values and gene counts, were visualized using bar plots, allowing for the identification of potential disease linkages.

#### 4.3.2. Cell-Type Specific Enrichment Analysis

To explore the cellular context of the identified genes, we conducted cell-type specific enrichment analysis using WebCSEA (Web-based Cell-type Specific Enrichment Analysis) [60]. This tool is designed to identify the enrichment of gene sets within specific human cell types by utilizing large-scale single-cell RNA sequencing data across multiple human tissues.

WebCSEA provides a comprehensive database of gene expression signatures from 1355 human cell types derived from 61 adult and fetal tissues. Enrichment was determined via permutation-based statistics. This analysis provided insights into the cellular environments most associated with the identified genes, offering a deeper understanding of their potential biological roles.

#### 4.3.3. Drug Target Enrichment Analysis

To identify potential therapeutic targets among the genes associated with SjD, we performed a drug target enrichment analysis using data from the Therapeutic Target Database (TTD). The TTD provides comprehensive information on therapeutic molecular targets, associated diseases, and corresponding medications, including their experimental status [20].

The enrichment analysis was conducted using our custom TargetEnrich.R script (https://github.com/astrika/Analytical-Approach-for-MS-GRN, accessed on 15 November 2024). We collected data on drug targets and related drug information directly from the TTD. The enrichment level of drug targets was assessed using a hypergeometric test, which evaluated the overlap between the drug target genes and our gene list. In the analysis, we considered 2578 drug target genes retrieved from the TTD.

### 4.4. Genetic Correlation Analysis

To assess the genetic overlap between SjD and other traits, we employed linkage disequilibrium score regression (LDSC) [61], a method that utilizes GWAS summary statistics to account for linkage disequilibrium (LD) patterns. LDSC works by regressing squared GWAS test statistics on LD scores, utilizing reference data from the 1000 Genomes Project to control for LD structure. We used LD information based on European ancestry.

In this analysis, we applied bivariate LDSC to estimate the genetic correlations (rg) between traits, which also adjusts for potential confounding factors such as population stratification. This method models the relationship between the Z-scores of the two traits, providing robust estimates of their shared genetic architecture. The LDSC software (v1.0.1) was used for this analysis, and statistical significance was set at *p* < 0.05. We calculated the rg of SjD with other AIDs, cancer, ocular, and aging diseases. The disease-specific GWAS datasets were sourced from the NHGRI-EBI GWAS Catalog and the GWAS Atlas (https://atlas.ctglab.nl/, accessed on 15 November 2024) [11,62,63,64,65,66,67,68,69,70,71,72,73,74]. This analysis provided insights into shared genetic architecture between SjD and related diseases, helping to identify potential common genetic risk factors across multiple conditions.

### 4.5. Mendelian Randomization Analysis

To investigate causal relationships between SjD and other traits, we performed Mendelian randomization (MR) analysis using the two-sample MR (2SMR) package (v0.6.5) [75]. MR utilizes genetic variants as instrumental variables (IVs) to estimate causal effects, minimizing confounding and reverse causation biases inherent in observational studies. The approach assumes that these genetic variants influence the outcome only through their effect on the exposure trait.

In this study, we applied two-sample MR, where the association between genetic variants and the exposure is estimated in one genome-wide association study (GWAS), and the association with the outcome is obtained from a separate GWAS. Using 2SMR, we selected single-nucleotide polymorphisms (SNPs) significantly associated with the exposure trait as instrumental variables, ensuring minimal linkage disequilibrium (LD) between SNPs through clumping procedures.

To assess the causality, we employed the inverse-variance weighted (IVW) regression as the primary MR method, which provides an unbiased causal estimate under the assumption that all SNPs are valid IVs. Sensitivity analyses were conducted using methods such as MR-Egger and weighted median to account for potential horizontal pleiotropy and confirm the robustness of our findings. Statistical significance was set at a *p*-value < 0.05.

## 5. Conclusions

In summary, this study presents a comprehensive multi-omics investigation of SjD, identifying key genetic networks and pathways involved in autoimmunity and cancer. By integrating genetic, epigenomic, and transcriptomic data, we provide novel insights into SjD pathogenesis, genetic correlations with other diseases, and potential therapeutic targets. Our findings demonstrate the power of network-based multi-omics integration in complex disease research and highlight drug repositioning opportunities for SjD. Future studies should focus on validating these candidate genes and pathways in experimental models to advance personalized therapeutic strategies for SjD patients.

## Figures and Tables

**Figure 1 ijms-26-04637-f001:**
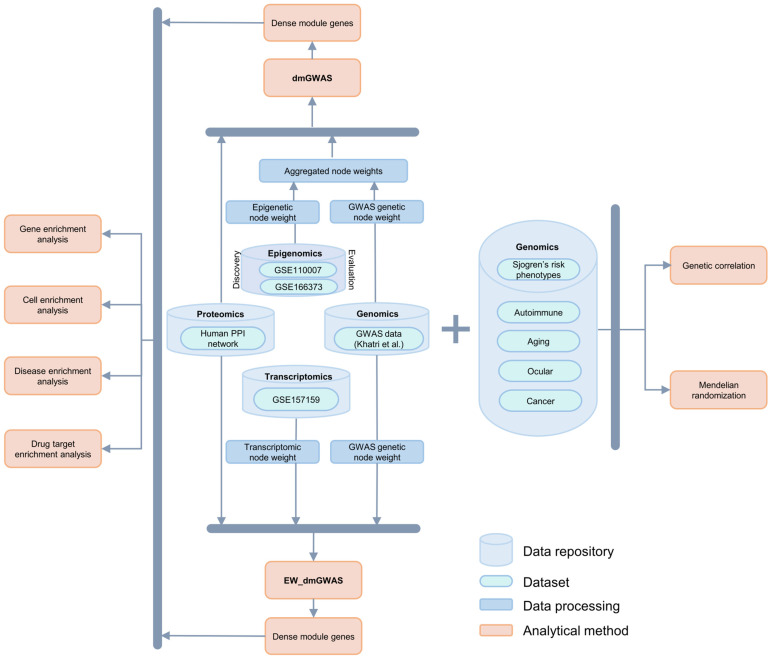
Workflow of the study design. We integrated the epigenomic and transcriptomic data with the GWAS summary statistics using network-based analysis to identify the top gene networks associated with SjD. We further selected the top gene networks for downstream analysis. We also compared the genetic and causal relationship of SjD with other diseases.

**Figure 2 ijms-26-04637-f002:**
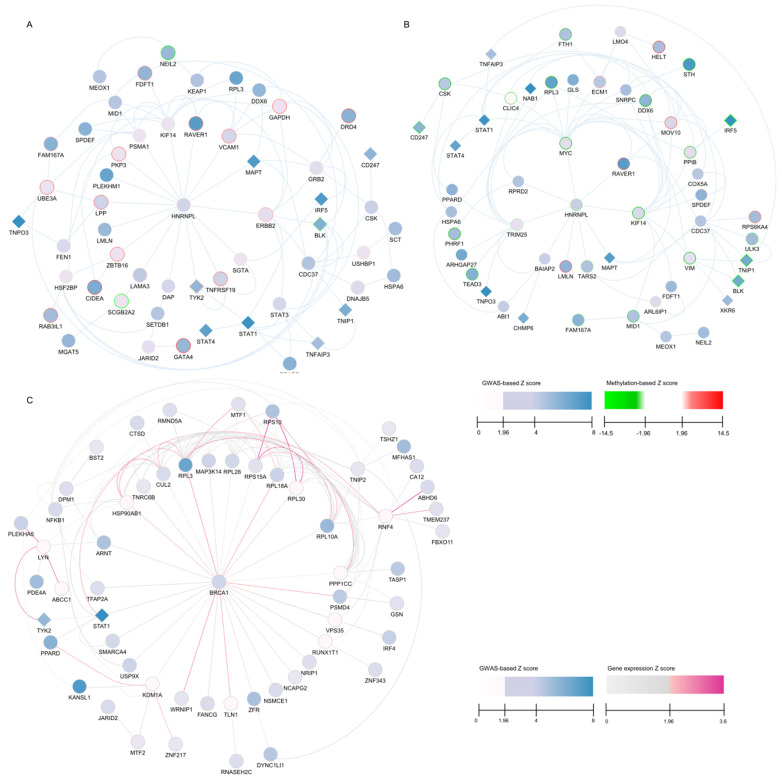
Protein—protein interaction (PPI) subnetworks across the selected top gene modules: (**A**) PPI network of the dmGWAS module genes using the discovery data; (**B**) PPI network of the dmGWAS module genes using the evaluation data; (**C**) PPI network of EW_dmGWAS module genes.

**Figure 3 ijms-26-04637-f003:**
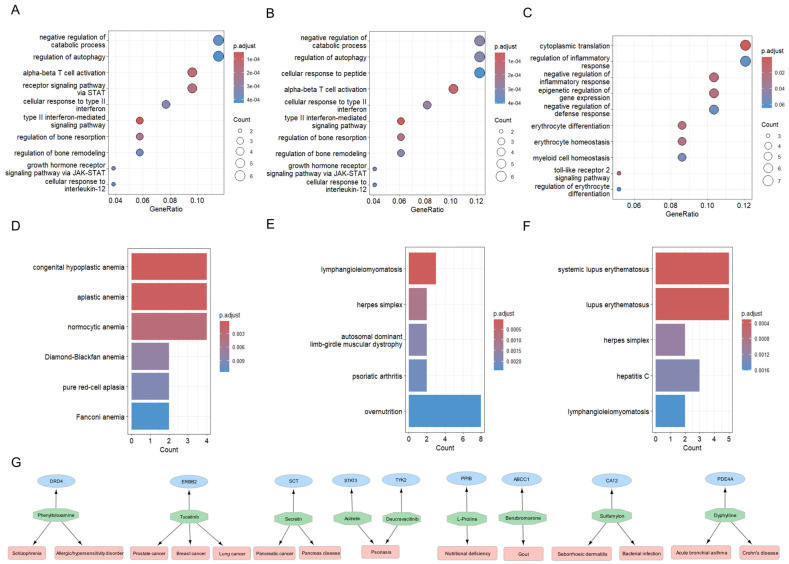
Over—representation analysis (ORA), disease—enrichment analysis (DEA), and drug—target enrichment analysis (DTEA) on the selected top gene modules: (**A**) ORA of the top gene modules from the discovery data based on dmGWAS analysis; (**B**) ORA of the top gene modules from the evaluation data based on dmGWAS analysis; (**C**) ORA of the top gene modules from the transcriptomic data based on the EW_dmGWAS analysis; (**D**) DTEA of the top gene modules from the discovery data based on dmGWAS analysis; (**E**) DTEA of the top gene modules from the discovery data based on dmGWAS analysis; (**F**) ORA of the top gene modules from the transcriptomic data based on the EW_dmGWAS analysis; (**G**) DTEA of epigenomic and transcriptomic genes.

**Figure 4 ijms-26-04637-f004:**
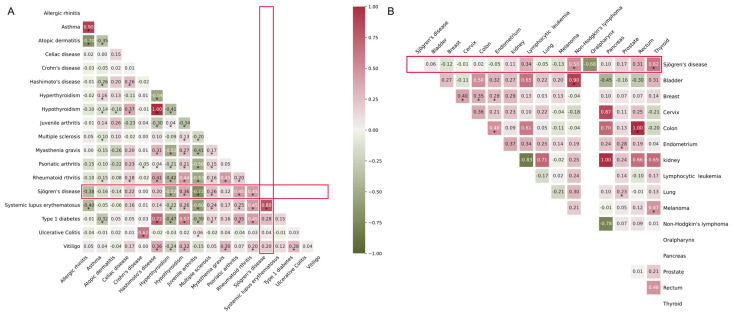
Heatmaps of genetic correlation analyses: (**A**) genetic correlation of SjD and 17 other autoimmune diseases (AIDs); (**B**) genetic correlation of SjD with 16 cancer types. *—denotes the diseases are statistically significant at *p*-value < 0.05. Highlighted red rectangle bars show the genetic correlation of SjD with 17 other AIDs.

**Figure 5 ijms-26-04637-f005:**
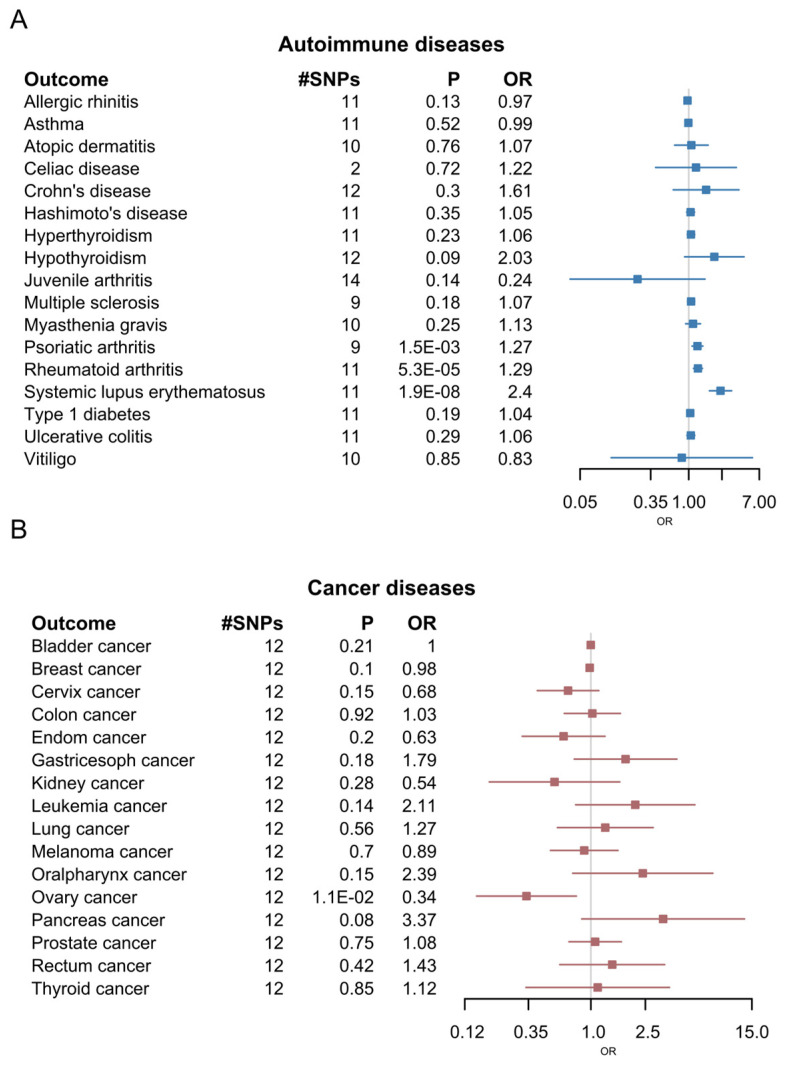
Forest plot of Mendelian randomization analysis: (**A**) Mendelian randomization of SjD with 17 other autoimmune diseases (AIDs); (**B**) Mendelian randomization of SjD with 16 cancer types.

## Data Availability

The SjD genome-wide association studies (GWAS) dataset is accessible at the dbGaP (phs000672.v1p1) with approved use (approved project #32032). Other datasets and tools used in this study are publicly available.

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
