# Peer review of "Genetic, Transcriptomic, and Epigenomic Insights into Sjögren’s Disease: An Integrative Network Investigation and Immune Diseases Comparison"

_ijms, 2025, doi:10.3390/ijms26104637_

Round 1
Reviewer 1 Report
Comments and Suggestions for Authors
The study explores the genetic, transcriptomic, and epigenomic profiles of Sjögren's disease, uncovering pivotal gene networks linked to autoimmune processes and cancer. Through integrative multi-omics analysis, it reveals common molecular mechanisms between Sjögren's disease and other immune-related conditions, suggesting potential biomarkers and drug targets for repurposing.
Although this article is interesting, I have some requests:
It would be interesting to investigate whether genetic modules identified in this study overlap with known antiphospholipid antibody syndrome-related pathways.
The manuscript doesn't mention if there are differences in gene expression and network modules between male and female patients. Since autoimmune diseases often affect men and women differently, it's important to check if certain pathways are regulated differently in each sex. Were the analyses done separately for men and women?
Again, Could you explore the relationship between sex differences, adding some references about this aspect in autoimmune diseases, particularly in antiphospholipid antibody syndrome and rheumatoid arthritis... ?
I would also like the authors to discuss what may happen in patients with X chromosome aneuploidies. Indeed, recent studies have explored autoimmunity and inflammation in patients with Klinefelter syndrome.
The Wnt signaling pathway is crucial for immune regulation and tissue homeostasis. However, its role in Sjögren's Disease is not discussed. Were any Wnt-related genes or pathways enriched in the identified gene modules? This should be explored, as Wnt signaling has been implicated in other autoimmune and inflammatory conditions.
Author Response
Comment 1:
The study explores the genetic, transcriptomic, and epigenomic profiles of Sjögren's disease, uncovering pivotal gene networks linked to autoimmune processes and cancer. Through integrative multi-omics analysis, it reveals common molecular mechanisms between Sjögren's disease and other immune-related conditions, suggesting potential biomarkers and drug targets for repurposing.
Although this article is interesting, I have some requests:
It would be interesting to investigate whether genetic modules identified in this study overlap with known antiphospholipid antibody syndrome-related pathways.
Response 1:
We thank the reviewer very much for the time on carefully evaluating our manuscript and providing us valuable comments. Please see our responses below. In revision, we discussed the top gene module relationship with antiphospholipid antibody syndrome (APS) related pathways. We added a paragraph in the Discussion section, copied below.
Line 312: “We further explored if the TMGs of SjD are associated with the antiphospholipid antibody syndrome (APS) pathways. The presence of antiphospholipid antibodies is part of the diagnostic criteria for several rheumatic diseases, including Sjogren’s disease, and often leads to serious thrombotic events [1, 2]. Several TMGs have been implicated in immune regulation, signaling, and cellular stress pathways relevant to APS (BLK, CDC37, DDX6, FAM167A, STAT1 and STAT4). STAT1 and STAT4 are directly involved in interferon and cytokine signaling, which are central to APS-related inflammation and autoimmunity [3, 4]. BLK and FAM167A influence B-cell function and autoantibody production, contributing to the generation of pathogenic antiphospholipid antibodies [4, 5]. DDX6 and CDC37 regulate cellular stress responses and autophagy, processes that can impact endothelial function and thrombosis in APS [6, 7]. While MAPT, TNIP1, TNPO3, and UBE3A have less clearly defined roles in APS.”
Comment 2:
The manuscript doesn't mention if there are differences in gene expression and network modules between male and female patients. Since autoimmune diseases often affect men and women differently, it's important to check if certain pathways are regulated differently in each sex. Were the analyses done separately for men and women?
Response 2:
We thank the reviewer for pointing out the potential sex-based differences in gene expression and network modules. This is a very important research topic, especially for autoimmune disease. In our manuscript, we used GWAS data that included both males and females. However, the transcriptomic and epigenetic data were limited to female samples only. Due to not large sample size in GWAS studies, and overall lack of the omics data for Sjögren's disease, it will be difficult for us to explore sex difference at this stage. We will keep our eyes open on the future data release and study the genomic features in sex difference in future. We have added this as a limitation in the revised manuscript to clarify this point.
Line 362 “Third, our multi-omics analysis, which integrated GWAS with transcriptomic and epigenetic data, was conducted using datasets derived solely from females. This lack of gender diversity may limit the generalizability of our findings across both sexes. Future studies incorporating male data would be valuable to assess potential sex-specific differences in the molecular mechanisms underlying.”
Comment 3:
Again, Could you explore the relationship between sex differences, adding some references about this aspect in autoimmune diseases, particularly in antiphospholipid antibody syndrome and rheumatoid arthritis... ?
Response 3:
Thank you for your valuable comment. We agree that exploring sex differences in autoimmune diseases (AIDs) is very important. We added a paragraph in the Discussion section.
Line 325: “AIDs exhibit notable sex-based dimorphism, with a higher prevalence in females. This sex bias is particularly evident in SjD, APS and RA [8]. Male and female patients with SjD demonstrate distinct disease manifestations. SjD, for instance, has a striking female predominance, with female-to-male ratios ranging from 6:1 to 16:1, particularly in older women post-menopause [9]. APS also shows a female bias, with women typically developing the disease earlier and experiencing more venous thromboses, while men are more prone to arterial events [10]. RA affects women 2-3 times more often than men, with women often experiencing more severe symptoms and poorer treatment outcomes [11]. Females with SjD, are at higher risk for conditions like fibromyalgia, migraines, and Ehlers-Danlos syndrome, while males have increased risks of cerebrovascular and cardiac issues. However, the relationship between these comorbidities and SjD itself remains unclear [9]. These sex differences are believed to arise from complex interactions between sex hormones, genetics, and environmental factors, highlighting the need for sex-specific approaches in both research and treatment.”
Comment 4:
I would also like the authors to discuss what may happen in patients with X chromosome aneuploidies. Indeed, recent studies have explored autoimmunity and inflammation in patients with Klinefelter syndrome.
Response 4:
We sincerely thank the reviewer for this important comment. We appreciated the suggestion to discuss the potential implications of X chromosome aneuploidies. We added a paragraph in the Discussion section.
Line 339: “Further, the X chromosome plays a key role in the sex differences seen in SjD, as many immune response genes are located on the X chromosome. Females, with two X chromosomes, may be more prone to immune dysregulation than males, who have only one X chromosome [12]. A notable study found that women with trisomy X have a 2.9-fold higher risk of developing primary Sjögren's disease compared to women with normal XX karyotypes [8]. X chromosome aneuploidies, such as Klinefelter syndrome (47,XXY) in men and triple X syndrome (47,XXX) in women, are significantly enriched among patients with Sjögren’s syndrome compared to the general population, supporting the concept of an X chromosome dose effect in disease susceptibility [13, 14]. This increased risk is thought to be mediated by immune-related genes on the X chromosome—such as TLR7 and CXorf21—that escape X inactivation, leading to overexpression and heightened immune activation, which helps explain the strong female predominance observed in Sjögren’s syndrome [15]. Many of the significant genetic associations related to SjD are concentrated in the MHC region and vary depending on the patient’s serological status [12]. This highlights the complex interplay between genetic factors, X chromosome dosage, and immune response in the development of SjD, suggesting that the female bias in this disease may be partly driven by X-linked gene expression. Currently, there has been no GWAS study specifically focused on sex difference in SjD, due to limited sample size in the existing studies.”
Comment 5:
The Wnt signaling pathway is crucial for immune regulation and tissue homeostasis. However, its role in Sjögren's Disease is not discussed. Were any Wnt-related genes or pathways enriched in the identified gene modules? This should be explored, as Wnt signaling has been implicated in other autoimmune and inflammatory conditions.
Response 5:
We thank the reviewer for this valuable comment on Wnt pathway. Our network-based approaches did not identify any genes associated with the Wnt signaling pathway in the gene modules we analyzed. While Wnt signaling is indeed important in immune regulation and has been implicated in other autoimmune conditions, it was not found to be enriched in this study’s results. One possible reason is that this pathway is relatively small, and our results relies on the genetic association signals.
References:
- Asherson RA, Fei HM, Staub HL, Khamashta MA, Hughes GR, Fox RI. Antiphospholipid antibodies and HLA associations in primary Sjogren's syndrome. Ann Rheum Dis. 1992;51(4):495-8. Epub 1992/04/01. doi: 10.1136/ard.51.4.495. PubMed PMID: 1586247; PubMed Central PMCID: PMCPMC1004699.
- Grygiel-Gorniak B, Mazurkiewicz L. Positive antiphospholipid antibodies: observation or treatment? J Thromb Thrombolysis. 2023;56(2):301-14. Epub 2023/06/02. doi: 10.1007/s11239-023-02834-6. PubMed PMID: 37264223; PubMed Central PMCID: PMCPMC10234248.
- Najjar I, Fagard R. STAT1 and pathogens, not a friendly relationship. Biochimie. 2010;92(5):425-44. Epub 2010/02/18. doi: 10.1016/j.biochi.2010.02.009. PubMed PMID: 20159032; PubMed Central PMCID: PMCPMC7117016.
- Yin H, Borghi MO, Delgado‐Vega AM, Tincani A, Meroni PL, Alarcón‐Riquelme ME. Association of STAT4 and BLK, but not BANK1 or IRF5, with primary antiphospholipid syndrome. Arthritis & Rheumatism. 2009;60(8):2468-71. doi: 10.1002/art.24701.
- Barinotti A, Radin M, Cecchi I, Foddai SG, Rubini E, Roccatello D, et al. Genetic Factors in Antiphospholipid Syndrome: Preliminary Experience with Whole Exome Sequencing. Int J Mol Sci. 2020;21(24). Epub 2020/12/19. doi: 10.3390/ijms21249551. PubMed PMID: 33333988; PubMed Central PMCID: PMCPMC7765384.
- Xu X, Wang J, Zhang Y, Yan Y, Liu Y, Shi X, et al. Inhibition of DDX6 enhances autophagy and alleviates endoplasmic reticulum stress in Vero cells under PEDV infection. Vet Microbiol. 2022;266:109350. Epub 2022/01/28. doi: 10.1016/j.vetmic.2022.109350. PubMed PMID: 35085948.
- Tang Z, Shi H, Chen C, Teng J, Dai J, Ouyang X, et al. Activation of Platelet mTORC2/Akt Pathway by Anti-β2GP1 Antibody Promotes Thrombosis in Antiphospholipid Syndrome. Arteriosclerosis, Thrombosis, and Vascular Biology. 2023;43(10):1818-32. doi: 10.1161/atvbaha.123.318978.
- Punnanitinont A, Kramer JM. Sex-specific differences in primary Sjögren's disease. Frontiers in Dental Medicine. 2023;4. doi: 10.3389/fdmed.2023.1168645.
- Bruno KA, Morales-Lara AC, Bittencourt EB, Siddiqui H, Bommarito G, Patel J, et al. Sex differences in comorbidities associated with Sjogren's disease. Front Med (Lausanne). 2022;9:958670. Epub 2022/08/23. doi: 10.3389/fmed.2022.958670. PubMed PMID: 35991633; PubMed Central PMCID: PMCPMC9387724.
- Truglia S, Capozzi A, Mancuso S, Manganelli V, Rapino L, Riitano G, et al. Relationship Between Gender Differences and Clinical Outcome in Patients With the Antiphospholipid Syndrome. Front Immunol. 2022;13:932181. Epub 2022/07/22. doi: 10.3389/fimmu.2022.932181. PubMed PMID: 35860235; PubMed Central PMCID: PMCPMC9289158.
- Fairweather D, Beetler DJ, McCabe EJ, Lieberman SM. Mechanisms underlying sex differences in autoimmunity. Journal of Clinical Investigation. 2024;134(18). doi: 10.1172/jci180076.
- Imgenberg-Kreuz J, Rasmussen A, Sivils K, Nordmark G. Genetics and epigenetics in primary Sjögren’s syndrome. Rheumatology. 2021;60(5):2085-98. doi: 10.1093/rheumatology/key330.
- Mougeot JL, Noll BD, Bahrani Mougeot FK. Sjögren's syndrome X‐chromosome dose effect: An epigenetic perspective. Oral Diseases. 2018;25(2):372-84. doi: 10.1111/odi.12825.
- Harris VM, Sharma R, Cavett J, Kurien BT, Liu K, Koelsch KA, et al. Klinefelter's syndrome (47,XXY) is in excess among men with Sjögren's syndrome. Clinical Immunology. 2016;168:25-9. doi: 10.1016/j.clim.2016.04.002.
- Sharma R, Harris VM, Cavett J, Kurien BT, Liu K, Koelsch KA, et al. Brief Report: Rare X Chromosome Abnormalities in Systemic Lupus Erythematosus and Sjögren's Syndrome. Arthritis & Rheumatology. 2017;69(11):2187-92. doi: 10.1002/art.40207.
Reviewer 2 Report
Comments and Suggestions for Authors
Detailed review of the document uploaded to word.

Author Response
Comment 1:
In the article, American scientists studied genetic, transcriptomic and epigenomic data in Sjogren's disease, because there are no exact mechanisms of the development of this disease. For this study, intra-network approaches and dense modules of GWAS and dmGWAS were used to link their functional role of investigating associations with autoimmune and oncological diseases. 10 and 12 genes were selected for the modules, and 5 and 7 significant genes were selected from them. These genes reinforce the genetic links between Sjögren's syndrome and autoimmune and oncological diseases, which can cause susceptibility and progression of the disease. The cross-over between open and assessment data was also assessed. selection and visualization of genetic data was carried out, biological mechanisms were investigated in the analysis of gene enrichment, diseases and cellular specificity. Through the analysis of target drug enrichment, TMG identified 31 genes encoding target drugs that are associated with various types of cancer and autoimmune diseases, including drugs for the treatment of Sjogren's syndrome - tucatinib, secretin, phenyltoloxamine, acitretin. Sjogren's syndrome had a strong correlation with clinical manifestations of autoimmune diseases, lymphoma, non-Hodgkin's lymphoma, cancer of the oral cavity, thyroid gland, ovaries, psoriasis, RA, SLE, which was reliably proven.
Therefore, in this study, multiomics was used to investigate genetic data, which allowed to model genetic links between Sjogren's syndrome and other diseases through immune response pathways, such as interferon signaling, STAT 1, CDC37, DDX6 MAPT, RPL-3, T-lymphocyte activation. Modern statistical data based on multiomics, including genetic correlation analysis, Mendelian randomized analysis, regression with inverse variance, were used for statistical evaluation. This modern research has good results and the prospect of further research.
Among the 60 sources of literature, 24 sources are 5 years old, in the future it is desirable that the sources are no older than 5 years old. In general, I recommend this article for publication in the journal Мolecular Sciences.
Response 1:
We thank the reviewer for very thoughtful comments and positive evaluation of our manuscript. We appreciate your recommendation for publication and have carefully considered your suggestions, including updating the references to ensure more recent literature is included. Your feedback has helped us improve the clarity and quality of the manuscript.
Reviewer 3 Report
Comments and Suggestions for Authors
In this work the authors have conducted a complex bioinformatic analysis on Sjögren's disease datasets. The genetic, epigenetic, transcriptomic, proteomic analysis have highlighted some potentially targetable pathways, also by some repurposable drugs. The discussion is convincing, and the limits of the study are correctly exposed.
Author Response
Comment 1:
In this work the authors have conducted a complex bioinformatic analysis on Sjögren's disease datasets. The genetic, epigenetic, transcriptomic, proteomic analysis have highlighted some potentially targetable pathways, also by some repurposable drugs. The discussion is convincing, and the limits of the study are correctly exposed.
Response 1:
We thank the reviewer for the positive and encouraging feedback. We are pleased that you found our bioinformatic analysis and discussion convincing, and we appreciate your recognition of the study’s strengths, including the identification of potentially targetable pathways and drug repurposing opportunities. Your acknowledgment of our effort to clearly outline the study’s limitations is also greatly appreciated.
Reviewer 4 Report
Comments and Suggestions for Authors
1. Scientific Quality and Originality
The manuscript presents an ambitious, innovative, and well-designed analysis of the molecular basis of Sjögren’s disease (SjD), integrating genetic (GWAS), epigenetic (DNA methylation), and transcriptomic data. The use of two network-based approaches (dmGWAS and EW_dmGWAS) allows for the identification of genes and pathways relevant to SjD pathogenesis and their connections with other autoimmune diseases and cancers.
Strengths:
- Innovative network-based approach to multi-layered data integration.
- High quality and depth of bioinformatics analyses.
- The use of Mendelian randomization and genetic correlation analysis adds translational value to the findings.
2. Methodology and Data Quality
The authors applied well-established bioinformatic methods, complemented by their own solutions (e.g., scaling of weights in gene networks). The methodology is described in detail and with transparency. The division into discovery and validation datasets is a significant strength that enhances the robustness of the results.
3. Clarity and Structure
The manuscript is clearly and logically written. Each section guides the reader step-by-step through the analytical process. The abstract provides a concise summary of the study's aim, methodology, results, and significance.
4. Figures and Tables
Figures are well-prepared and correspond appropriately to the described results.
5. References
The reference list is extensive and up to date. The authors cite key studies related to autoimmune disease genomics, omics data integration, and network analysis.
6. Weaknesses and Areas for Improvement
Although there are a few limitations—such as the use of minor salivary glands only and a lack of ethnic diversity in the datasets—the authors acknowledge these in the manuscript.
Author Response
Comment 1:
- Scientific Quality and Originality
The manuscript presents an ambitious, innovative, and well-designed analysis of the molecular basis of Sjögren’s disease (SjD), integrating genetic (GWAS), epigenetic (DNA methylation), and transcriptomic data. The use of two network-based approaches (dmGWAS and EW_dmGWAS) allows for the identification of genes and pathways relevant to SjD pathogenesis and their connections with other autoimmune diseases and cancers.
Strengths:
- Innovative network-based approach to multi-layered data integration.
- High quality and depth of bioinformatics analyses.
- The use of Mendelian randomization and genetic correlation analysis adds translational value to the findings.
- Methodology and Data Quality
The authors applied well-established bioinformatic methods, complemented by their own solutions (e.g., scaling of weights in gene networks). The methodology is described in detail and with transparency. The division into discovery and validation datasets is a significant strength that enhances the robustness of the results.
- Clarity and Structure
The manuscript is clearly and logically written. Each section guides the reader step-by-step through the analytical process. The abstract provides a concise summary of the study's aim, methodology, results, and significance.
- Figures and Tables
Figures are well-prepared and correspond appropriately to the described results.
- References
The reference list is extensive and up to date. The authors cite key studies related to autoimmune disease genomics, omics data integration, and network analysis.
- Weaknesses and Areas for Improvement
Although there are a few limitations—such as the use of minor salivary glands only and a lack of ethnic diversity in the datasets—the authors acknowledge these in the manuscript.
Response 1:
We thank the reviewer for the thoughtful and detailed review. We greatly appreciate your recognition of the innovative approach, the depth of our bioinformatics analysis, and the translational value of our methodology. We are also pleased that the clarity and structure of the manuscript met your expectations.
Round 2
Reviewer 1 Report
Comments and Suggestions for Authors
The manuscript is ready fo publication